# Interrogating basal ganglia circuit function in people with Parkinson's disease and dystonia

Srdjan Sumarac[1,2†], Kiah A Spencer[1,2†], Leon A Steiner[2,3,4†], Conor Fearon[2,5,6], Emily A Haniff[2], Andrea A Kühn[3], Mojgan Hodaie[2,7,8,9], Suneil K Kalia[2,8,9,10], Andres Lozano[2,7,8,9], Alfonso Fasano[2,5,6,7,8], William Duncan Hutchison[2,8,9,11], Luka Milosevic[1,2,7,8,10]*

[1]Institute of Biomedical Engineering, University of Toronto, Toronto, Canada; [2]Krembil Brain Institute, University Health Network, Toronto, Canada; [3]Department of Neurology and Experimental Neurology, Charité-Universitätsmedizin Berlin, Berlin, Germany; [4]Berlin Institute of Health (BIH), Berlin, Germany; [5]Edmond J. Safra Program in Parkinson's Disease, Morton and Gloria Shulman Movement Disorders Clinic, Toronto Western Hospital, Toronto, Canada; [6]Department of Neurology, University of Toronto, Toronto, Canada; [7]Institute of Medical Sciences, University of Toronto, Toronto, Canada; [8]Center for Advancing Neurotechnological Innovation to Application (CRANIA), Toronto, Canada; [9]Department of Surgery, University of Toronto, Toronto, Canada; [10]KITE, University Health Network, Toronto, Canada; [11]Department of Physiology, University of Toronto, Toronto, Canada

*For correspondence: luka.milosevic@mail.utoronto.ca

†These authors contributed equally to this work

Competing interest: The authors declare that no competing interests exist.

## Abstract

**Background:** The dichotomy between the hypo- versus hyperkinetic nature of Parkinson's disease (PD) and dystonia, respectively, is thought to be reflected in the underlying basal ganglia pathophysiology. In this study, we investigated differences in globus pallidus internus (GPi) neuronal activity, and short- and long-term plasticity of direct pathway projections.

**Methods:** Using microelectrode recording data collected from the GPi during deep brain stimulation surgery, we compared neuronal spiketrain features between people with PD and those with dystonia, as well as correlated neuronal features with respective clinical scores. Additionally, we characterized and compared readouts of short- and long-term synaptic plasticity using measures of inhibitory evoked field potentials.

**Results:** GPi neurons were slower, burstier, and less regular in dystonia. In PD, symptom severity positively correlated with the power of low-beta frequency spiketrain oscillations. In dystonia, symptom severity negatively correlated with firing rate and positively correlated with neuronal variability and the power of theta frequency spiketrain oscillations. Dystonia was moreover associated with less long-term plasticity and slower synaptic depression.

**Conclusions:** We substantiated claims of hyper- versus hypofunctional GPi output in PD versus dystonia, and provided cellular-level validation of the pathological nature of theta and low-beta oscillations in respective disorders. Such circuit changes may be underlain by disease-related differences in plasticity of striato-pallidal synapses.

**Funding:** This project was made possible with the financial support of Health Canada through the Canada Brain Research Fund, an innovative partnership between the Government of Canada (through Health Canada) and Brain Canada, and of the Azrieli Foundation (LM), as well as a grant from the Banting Research Foundation in partnership with the Dystonia Medical Research Foundation (LM).

### eLife assessment

This is a **valuable** study of the responses of GPi neurons to deep brain stimulation (DBS) in human Parkinson disease and dystonia patients and finds **convincing** evidence for altered short-term and long-term plasticity in response to DBS between the two patient populations. This dataset is of interest to both basic and clinical researchers working in the field of DBS and movement disorders.

## Introduction

Parkinson's disease (PD) is a complex neurodegenerative disorder characterized by non-motor manifestations and motor symptoms, including tremor, bradykinesia, rigidity, postural instability, and freezing of gait (*Jankovic, 2008*). Dystonia is a movement disorder characterized by involuntary muscle contractions, repetitive movements, and abnormal fixed postures (*Albanese et al., 2013*). While PD is characterized as a collection of hypokinetic syndromes (apart from tremor and levodopa-induced dyskinesias), dystonia is considered a hyperkinetic movement disorder.

The dichotomy between the hypo- versus hyperkinetic nature of PD and dystonia, respectively, is thought to be reflected in the underlying basal ganglia pathophysiology. In particular, basal ganglia circuitry that involves 'direct' (D1-mediated striatal projections to the internal globus pallidus [GPi]) and 'indirect' (D2-mediated striatal projections via the external globus pallidus and subthalamic nucleus to GPi) pathways has been implicated in these disorders (*DeLong and Wichmann, 2007*). In PD, activity in the indirect pathway is upregulated while the direct pathway is downregulated, both of which converge toward increased GPi-mediated inhibition of thalamocortical motor networks (*DeLong, 1990*). Conversely, dystonia has been hypothesized to occur as a result of hypofunctional indirect pathway activity (*Naumann et al., 1998*; *Simonyan et al., 2013*) and/or hyperfunctional direct pathway activity (*Simonyan et al., 2017*), producing decreased GPi-mediated inhibition of thalamo-cortical motor networks. These circuit differences would thereby result in increased GPi firing rates (FRs) in PD and decreased rates in dystonia, which has been substantiated by some (*Starr et al., 2005*; *Tang et al., 2007*) but not all (*Hutchison et al., 2003*) human single-neuron studies. In addition to potential differences in single-neuron firing properties, each of the disorders has been associated with aberrant synchronization at the neural aggregate level; namely, increased beta (13–30 Hz) and theta (4–12 Hz) local field potential (LFP) oscillations in PD (*Neumann et al., 2016*) and dystonia (*Neumann et al., 2017*), respectively.

Despite opposing neurocircuit pathologies, deep brain stimulation (DBS) of the GPi is a widely used treatment for the motor symptoms of both PD and dystonia. However, the timescales of symptomatic improvement after GPi-DBS and the reappearance of symptoms upon stimulation cessation are markedly longer in dystonia (minutes–hours–days) compared to PD (seconds–minutes) (*Stefani et al., 2019*). These differences in timescales may be the result of disease-specific dynamics of various forms of synaptic plasticity (*Ruge et al., 2011a*). Intriguingly, high-frequency GPi stimulation has been shown to induce synaptic depression of inhibitory striato-pallidal (GPi) projections during stimulation (short-term plasticity effects), followed by an enduring potentiation of these projections after stimulation cessation (long-term plasticity-like effects) in both patients with PD (*Milosevic et al., 2019*) and dystonia (*Liu et al., 2012*). However, the dynamics of these plastic effects have not been directly compared between disorders.

In this work, we compared GPi single-neuron activities (rate and oscillation -based features) between PD and dystonia, and investigated the relationships between electrophysiological features and disease severity. Moreover, we investigated differences in the dynamics of short- and long-term plasticity of the direct pathway projections.

## Methods

### Patients and clinical scores

For the single-neuron feature analysis (*Figure 1*), data were collected from 19 patients with non-genetic dystonia (135 included segments; see *Supplementary file 1* for information on subtypes) and 44 patients with PD (222 segments) during awake microelectrode-guided GPi-DBS surgeries (*Hutchison et al., 1994*) after overnight withdrawal of anti-Parkinsonian medications in patients with

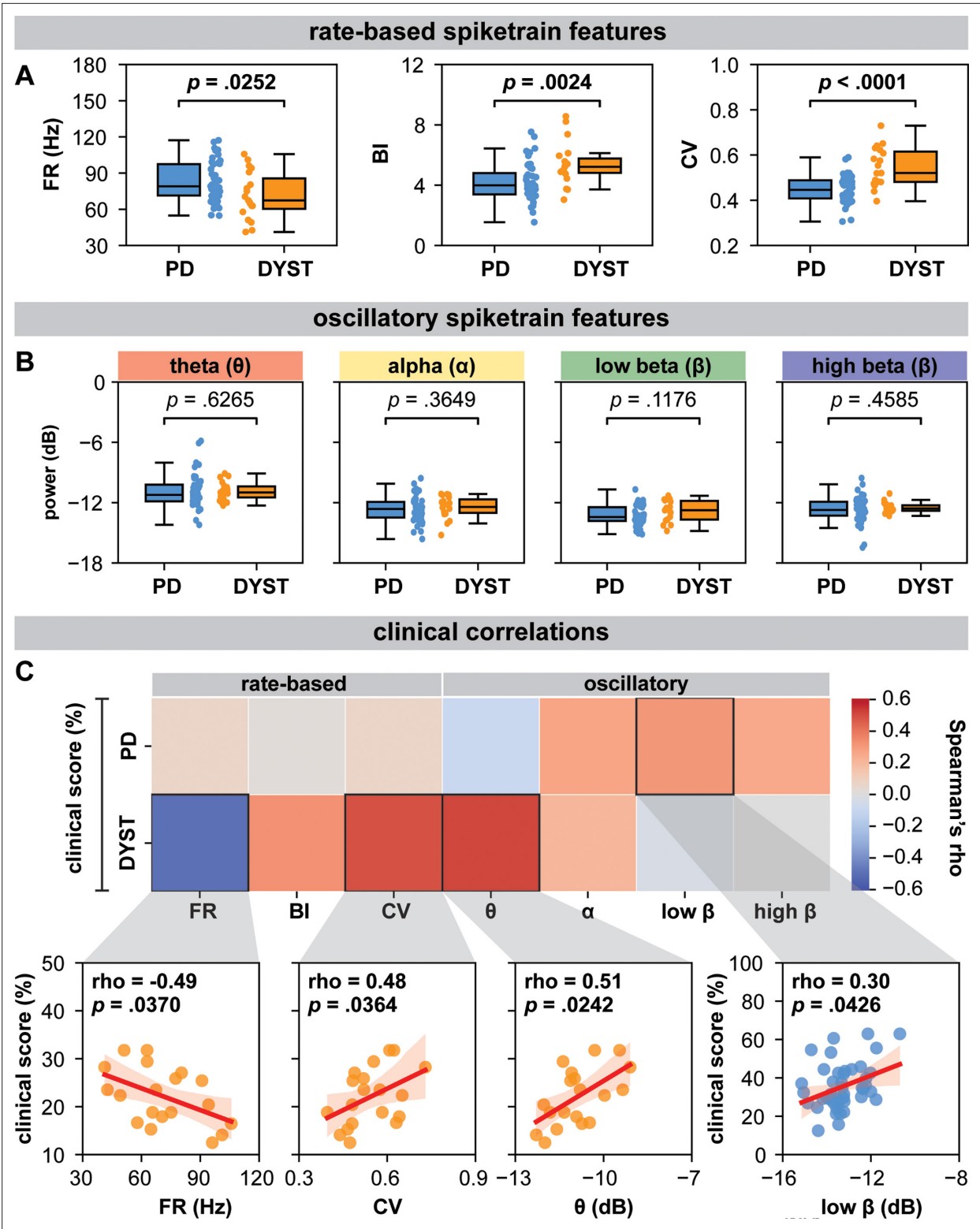

**Figure 1.** Globus pallidus internus (GPi) spiketrain feature analyses and clinical correlates of Parkinson's disease (PD) and dystonia. With respect to (**A**) rate-based spiketrain features, firing rate was greater in PD while burst index (BI) and coefficient of variation (CV) were greater in dystonia; whereas no differences were found for (**B**) oscillatory spiketrain features for theta, alpha, low-beta, and high-beta frequencies. Mann–Whitney U (MWU) statistical results depicted are not corrected for multiple comparisons; after correction using the Bonferroni method, only CV and BI results remain significant (see *Supplementary file 3a*). (**C**) In PD, the power of low-beta spiketrain oscillations positively correlated (Spearman correlation) with symptom severity; in dystonia, neuronal firing rate negatively correlated with symptom severity, whereas CV and the power of theta spiketrain oscillations positively correlated

*Figure 1 continued on next page*

*Figure 1 continued*

with symptom severity. Depicted scatterplots are results that were significant before correction for multiple comparisons; however, none of the results persist after Benjamini–Hochberg correction for false discovery rate (see *Supplementary file 3b*).

The online version of this article includes the following figure supplement(s) for figure 1:

**Figure supplement 1.** Rate-based and oscillatory spiketrain features across dystonia subtypes and Parkinson's disease (PD).

**Figure supplement 2.** Clinical correlations with Parkinson's disease (PD) hypokinetic symptoms.

PD. Corresponding preoperative clinical scores for PD using the Unified Parkinson's Disease Rating Scale Part III (UPDRSIII) and dystonia using either the Burke–Fahn–Marsden Dystonia Rating Scale (BFMDRS) or the Toronto Western Spasmodic Torticollis Rating Scale (TWSTRS) were also amalgamated for each patient. Given that UPDRSIII includes both hypokinetic and hyperkinetic symptoms of PD, we further sought to disaggregate the score by only considering items 23–26 in UPDRSIII, which assess hypokinetic symptoms of PD. To enable comparison across different dystonia scales, clinical scores were normalized (0–100%) with respect to the maximum severity per scale (108 for total UPDRSIII, 32 for hypokinetic only UPDRSIII, 85 for TWSTRS, and 120 for BFMDRS). For plasticity analyses (*Figure 2*), data were collected from a subset of 8 patients with dystonia (12 recording sites) and 10 patients with PD (13 recording sites). A data summary is available in *Supplementary file 1*. Each patient provided informed consent, including consent to publish, and experiments were approved by the University Health Network Research Ethics Board under the approval identifier 17-5006. The study adhered to the guidelines set by the tri-council Policy on Ethical Conduct for Research Involving Humans.

## Intraoperative data acquisition

Two independently driven microelectrodes (600 μm spacing; 0.2–0.4 MΩ impedances; ≥10 kHz; sampling rate) were used for recordings; amplified using two Guideline System GS3000 amplifiers (Axon Instruments, Union City, USA) and digitized using a CED1401 data acquisition system (Cambridge Electronic Design, Cambridge, UK). For single-neuron data, units were sampled across the spatial extent of GPi as part of the standard-of-care neurophysiological mapping procedure, as previously described (*Hutchison et al., 1994*). For plasticity data, microstimulation was delivered from one microelectrode using biphasic (cathodal followed by anodal) pulses (100 μA, 150 μs) from an isolated constant current stimulator (Neuro-Amp1A, Axon Instruments), while recording field-evoked potentials (fEPs) using an adjacent microelectrode at the same depth as done previously (*Milosevic et al., 2019*; *Liu et al., 2012*; *Milosevic et al., 2018*). Initially, 10 low-frequency stimulation (LFS) pulses were delivered at 1 Hz to obtain baseline measurements of fEP amplitudes (i.e., functional readouts striato-pallidal inhibitory efficacy). Then, a standard high-frequency stimulation (HFS) tetanizing protocol was delivered (four 2 s blocks of 100 Hz, each separated by 8 s) (*Liu et al., 2012*), after which another set of LFS pulses was delivered to obtain measurements of post-HFS fEP amplitudes (plasticity protocol depicted in *Figure 2A*). Long-term plasticity analysis involved quantifying fEP changes before versus after HFS. Short-term plasticity analysis involved quantification of successive fEP amplitudes during HFS.

## Offline analyses and statistics

GPi high-frequency discharge neurons (*Hutchison et al., 1994*) of >4 signal-to-noise ratio and <1% interspike interval violations were included in the single-neuron analyses. Segments were band-pass filtered (300–3000 Hz) and template matched, after which neurophysiological features were extracted from each segment, including per patient median FR, and the ninetieth percentile of the burst index (BI), coefficient of variation (CV), and oscillatory power across the theta (4–8 Hz), alpha (8–12 Hz), low-beta (12–21 Hz), and high-beta (21–30 Hz) frequency bands. Oscillatory power was extracted using Lomb's periodogram, performed on the autocorrelation function of single-neuron segments as previously described (*Kaneoke and Vitek, 1996*). The BI was computed by taking the ratio of the means from a two-component Gaussian mixture model applied to the log interspike interval distribution, a modification of the previous mode-over-mean ISI method (*Hutchison et al., 1994*). Neuronal features were compared between PD and dystonia using two-tailed Mann–Whitney *U* (MWU) tests due to non-normality of features. Spearman correlations were employed to investigate the relationships between

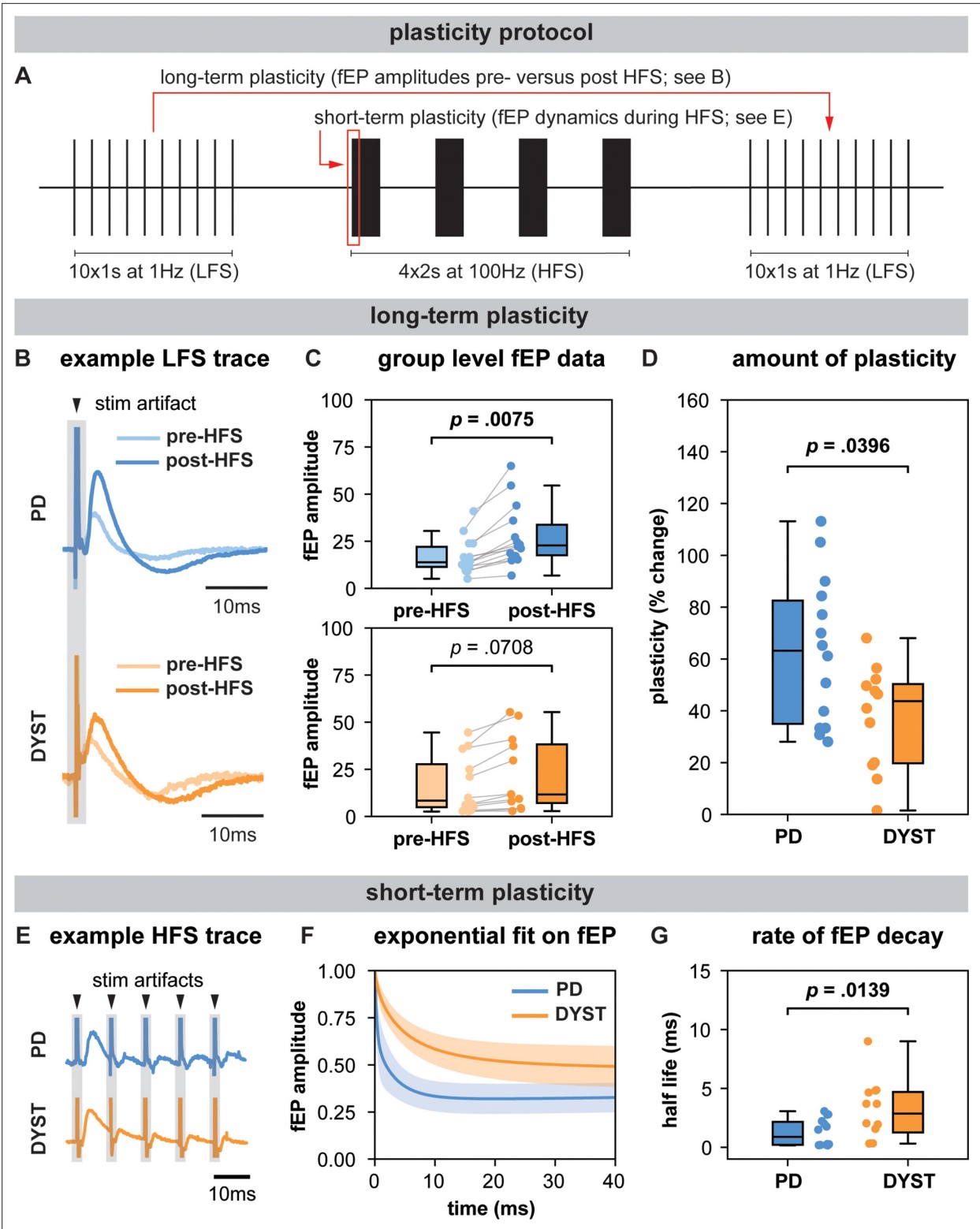

**Figure 2.** Long-term and short-term effects of high-frequency stimulation (HFS) on striato-pallidal plasticity in Parkinson's disease (PD) and dystonia. (**A**) Schematic of the plasticity protocol to assess long-term plasticity via field-evoked potential (fEP) amplitude comparisons pre- versus post-HFS and short-term plasticity via fEP dynamics during HFS. (**B**) highlights example fEP traces for measuring long-term plasticity pre- versus post-HFS, with (**C**) displaying group-level fEP amplitudes pre- versus post-HFS across diseases. (**D**) illustrates the amount of plasticity (i.e., percentage change in fEP amplitudes pre- versus post-HFS) in both PD and dystonia, with PD showing higher levels of plasticity. (**E**) provides an example of fEP traces during HFS

*Figure 2 continued on next page*

*Figure 2 continued*

for assessing short-term plasticity, with (**F**) depicting group-level decay rates of fEP amplitudes using an exponential fit on the fEP amplitudes over the first five stimulus pulses across diseases. (**G**) shows the half-life of the fitted exponential (i.e., rate of attenuation of fEP amplitudes) between PD and dystonia, with PD demonstrating faster fEP attenuation.

The online version of this article includes the following figure supplement(s) for figure 2:

**Figure supplement 1.** Long- and short-term plasticity across dystonia subtypes and Parkinson's disease (PD).

extracted features and clinical scores. For comparing differences in single-neuron features between PD and dystonia, significant results were followed up with post hoc multiple comparisons with a Bonferroni correction. For clinical correlations, non-parametric Monte Carlo permutation tests were used, avoiding assumptions about data distribution. The tested values were randomly shuffled 5000 times to form a probability distribution, with the p-value reflecting the original sample rank. All tests underwent adjustment for multiple comparisons, controlling the false discovery rate (FDR) at an α level of 0.05.

All fEP measurements for plasticity analyses were normalized with respect to the root mean square amplitude of the rectified pre-stimulation LFP (low-pass filtered at 50 Hz). For long-term plasticity analyses, the median normalized fEP amplitudes pre- versus post-HFS at each recording site were analyzed using a linear mixed model (LMM), with patient ID as a random factor, normalized fEP amplitudes as the response variable, and epoch as a fixed effect, to establish whether plasticity was elicited in each disorder. The percentage change from pre- to post-HFS was also compared across disorders using an LMM with patient ID as a random factor. For short-term plasticity analysis, the dynamic change of the first five fEP amplitudes during the first train of HFS was assessed, normalized with respect to the first fEP amplitude for each train. Given the relatively rapid rate of attenuation of fEP amplitudes in GPi with HFS (*Liu et al., 2012*; *Milosevic et al., 2018*), the first five fEPs were fit with a double-exponential function, and the half-life of the faster decaying exponential component was used as a readout of the rate of attenuation. Half-life measurements were then compared across disorders using an LMM with patient ID as a random factor.

## Results

### Differences in single-neuron features

In dystonia, FR was lower (MWU test; p=0.0252) and neurons were burstier (higher BI; MWU test; p=0.0024) and less regular (higher CV; MWU test; p<0.0001) compared to PD (*Figure 1A*). However, there were no significant differences in spiketrain oscillations in the delta (MWU test; p=0.6265), theta (MWU test; p=0.3649), low-beta (MWU test; p=0.1176), or high-beta (MWU test; p=0.4585) frequency bands (*Figure 1B*). The post hoc Bonferroni correction (adjusted significance threshold to p<0.0071) indicated that FR difference between PD and dystonia did not survive multiple comparisons (*Supplementary file 3*). A neuronal feature summary is available in *Supplementary file 2*.

### Clinical correlations

For rate-based spiketrain features, no clinical correlations were found for PD. In dystonia, symptom severity negatively correlated with FR (Spearman's rho = −0.49, p=0.0370) and positively correlated with CV (Spearman's rho = 0.48, p=0.0364; *Figure 1C*). For oscillatory spiketrain features, symptom severity positively correlated with low beta in PD (Spearman's rho = 0.30, p=0.0426), with a marginally stronger correlation for PD hypokinetic symptoms only (items 23–26 of UPDRSIII, Spearman's rho = 0.32, p=0.0330; *Figure 1—figure supplement 2*), and theta in dystonia (Spearman's rho = 0.51, p=0.0242; *Figure 1C*). However, none of the clinical correlations survived Benjamini–Hochberg FDR correction for multiple comparisons (*Supplementary file 3b*).

### Long-term plasticity-like effects

In PD, fEP amplitudes were significantly greater after compared to before HFS (LMM; p=0.0075, effect size = 5.42 ± 1.79; *Figure 2C*), while in dystonia, the increase approached but did not reach statistical significance (LMM; p=0.0708, effect size = 2.82 ± 1.45; *Figure 2C*). The percent change in fEP

amplitude was significantly greater in PD compared to dystonia (LMM; p=0.0396, effect size = 12.40 ± 5.41; *Figure 2D*).

### Short-term plasticity

Half-life values extracted from double exponential fits on the first five fEPs during HFS were significantly greater in dystonia compared to PD (LMM; p=0.0139, effect size = 1.04 ± 0.37; *Figure 2G*).

## Discussion

In this work, we leveraged the intraoperative environment to characterize GPi spiketrain features and to record and manipulate the efficacy of inhibitory direct pathway striato-pallidal projections in order to derive insights about disease-related features of basal ganglia circuit function in PD and dystonia. In particular, we found that GPi neurons exhibited lower FRs, but greater burstiness and variability in dystonia compared to PD (*Figure 1A*). While no differences were found in the power of spiketrain oscillations across disorders (*Figure 1B*), we found that PD symptom severity positively correlated with the power of low-beta frequency spiketrain oscillations, whereas dystonia symptom severity positively correlated with the power of theta frequency spiketrain oscillations (*Figure 1C*). Dystonia symptom severity moreover correlated negatively with FR and positively with neuronal variability. These results are discussed in greater detail with respect to previous literature in the section 'Neuronal correlates of PD and dystonia'. In response to electrical stimulation (protocol depicted in *Figure 2A*), we found significant increases in the amplitudes of positive-going stimulation-evoked field potential amplitudes (considered to reflect striato-pallidal synaptic strength, as exemplified in *Figure 2B*) before versus after HFS in both PD and dystonia (*Figure 2C*), with recording sites in PD exhibiting significantly greater increases (*Figure 2D*). While changes to evoked potential amplitude before versus after stimulation can be considered to be reflective of long-term plasticity (*Milosevic et al., 2019*; *Milosevic et al., 2018*), the dynamics of evoked potentials during HFS (as depicted in *Figure 2E*) can be considered reflective of short-term synaptic plasticity (*Milosevic et al., 2018*; *Steiner et al., 2022*). To this end, our findings are suggestive of faster latency synaptic depression in PD compared to dystonia (*Figure 2F/G*). Plasticity findings are discussed in greater detail in the section 'Direct pathway plasticity'.

### Neuronal correlates of PD and dystonia

The basal ganglia circuit models of PD and dystonia have been suggested to be driven by an imbalance between direct and indirect pathways (*Hallett, 1998*; *Hallett, 1993*). While hyperfunctionality of the indirect pathway and hypofunctionality of the direct pathway are associated with PD (*McGregor and Nelson, 2019*), the direct pathway has been suggested to be hyperfunctional in dystonia (*Simonyan et al., 2017*). To this end, our findings (*Figure 1A*) of lower FRs in dystonia compared to PD may further substantiate (*Starr et al., 2005*; *Tang et al., 2007*) such claims, whereby greater inhibition of GPi would produce decreased inhibitory output, and thus under-inhibition of thalamocortical motor networks, giving rise to hyperkinetic motor symptoms, whereas the opposite would be true in PD (*DeLong and Wichmann, 2007*). These claims are further substantiated by findings of an inverse relationship between FR and disease severity in dystonia (*Figure 1C*; rate-based features), although the opposite was not found in PD. Despite the lack of correlations with FR in PD, our findings seem to align with those of Muralidharan and colleagues (*Muralidharan et al., 2016*), who showed that GPi neuronal FRs may not directly correlate with motor severity but exhibit variability across the disease severity continuum in Parkinsonian non-human primates (initially increasing, then decreasing, then increasing again at mild, moderate, and severe disease manifestations, respectively). Thus, while GPi discharge rates may change in PD, such changes may not be reflected by linear relationships with motor sign development and progression. Indeed, variability in spike FRs in PD may be reflected in the considerable overlap in spiking activity between PD and dystonia (*Figure 1A*), with many dystonia patients exhibiting higher discharge rates compared to patients with PD. While differences in discharge rates were nevertheless observed between PD and dystonia, it may be that the combination of rate and pattern (reflected in the BI and CV) changes best differentiates the two disorders.

From a neural synchronization standpoint, we recently showed that pathological oscillations at the neural aggregate/LFP level are likely encoded by periodic oscillations in the FRs of single neurons

(*Scherer et al., 2022*). In the current work, we did not find significant differences in the power of spiketrain oscillations across disorders (*Figure 1B*). Although previous research has reported differences in the LFP power between PD and dystonia (*Silberstein et al., 2003*; *Wang et al., 2018*), a study in non-human primates found no such differences in single-neuron oscillatory strength (*Starr et al., 2005*), as reflected in our findings. However, despite a lack of difference in overall power across disorders, we were able to derive disease/frequency-specific relationships with respect to clinical scores (*Figure 1C*; oscillatory features). In particular, we provided evidence of relationships between the power of spiketrain oscillations in the theta and low-beta frequency bands and the severity of motor symptoms in dystonia and PD, respectively, providing a cellular-level validation of previous frequency-specific LFP clinical correlation studies in people with PD (beta oscillations) (*Neumann et al., 2016*) and dystonia (theta oscillations) (*Neumann et al., 2017*). Indeed, Muralidharan and colleagues (*Muralidharan et al., 2016*) also showed linear group-level relationships between low-beta frequency spiketrain oscillations and disease severity in Parkinsonian non-human primates, despite the lack of linear relationships with spike discharge rates (as discussed above).

## Direct pathway plasticity

We have previously shown that the amplitude of positive-going inhibitory fEPs is directly associated with the duration of neuronal inhibition *Milosevic et al., 2019*; *Steiner et al., 2021*, thus serving as a viable substrate of inhibitory synaptic transmission efficacy. These observations are supported by work in animals demonstrating the reversal of both positive-going extracellular fields and neuronal inhibition with the application of GABA antagonists (*Yoshida and Precht, 1971*; *Precht and Yoshida, 1971*). Importantly, we previously *Steiner et al., 2021* demonstrated that low-latency inhibitory fields recorded in the subthalamic nucleus (GPe-mediated) are far smaller in amplitude and are resilient during HFS (i.e., minimal synaptic depression), whereas high-latency inhibitory fields recorded in substantia nigra pars reticulata and GPi (*Liu et al., 2012*) (striatum-mediated) are far larger in amplitude and are subject to potent synaptic depression during HFS, thus, enabling electrophysiological discernment of these distinct inhibitory projections in humans.

To this effect, our findings may suggest disease-specific differences in mechanisms underlying both short- (*Prescott et al., 2013*) and long-term (*Ueki et al., 2006*; *Tamura et al., 2009*) forms of synaptic plasticity of direct pathway projections (*Figure 2*). Converging evidence from past animal and human studies suggests that dystonia is associated with impaired synaptic function and abnormal synaptic plasticity (*Chen and Udupa, 2009*; *Peterson et al., 2010*; *Quartarone and Pisani, 2011*). Compared to healthy controls, it has been shown that transcranial magnetic stimulation-induced motor-evoked potentials are hyperexcitable in dystonia *Chen, 2000*; *Siebner et al., 1999*, and somatosensory and motor cortical plasticity is greater (*Tamura et al., 2009*). Likewise, enhanced long-term potentiation at cortico-striatal synapses has been shown in rodent models of dystonia (*Köhling et al., 2004*; *Martella et al., 2009*). While our finding that long-term potentiation effects are greater in PD compared to dystonia (*Figure 2D*) is difficult to corroborate with this literature, one potential explanation can be that all of our patients with PD are long-term users of levodopa. We have previously shown that the intake of this antiparkinsonian dopaminergic medication leads to potent increases in the magnitude of direct pathway plasticity (*Milosevic et al., 2019*). Although patients are 12 hr withdrawn from anti-Parkinsonian medications for surgery, it could be that striato-pallidal synapses are nevertheless chronically over-sensitized from prolonged use of dopaminergic medication, which is a well-known hypothesis related to the manifestation of levodopa-induced dyskinesias (a hyperkinetic feature) in PD (*Calabresi et al., 2015*). Indeed, a lack of depotentiation of striato-pallidal projections has previously been observed in patients with levodopa-induced dyskinesias (*Prescott et al., 2014*). As such, excessive plasticity of these projections may corroborate hyperkinetic features of dystonia and levodopa-induced dyskinesias in PD.

In addition, the variation in the modulation of these striato-pallidal projections to electrical stimulation may also be reflective of differences in the mechanism by which DBS operates across PD and dystonia, despite a common stimulation target. Clinical studies in dystonia have shown that DBS leads to a more rapid improvement in the transient, dynamic muscle contractions (phasic components) of the disorder compared to the sustained, continuous muscle contractions (tonic or fixed components) (*Krauss, 2002*). The early improvement of the phasic component has been suggested to be associated with direct suppression of abnormal activity in the GPi (*Barow et al., 2014*), whereas the improvement

of tonic components has been associated with long-term potentiation-like synaptic plasticity in TMS studies (*Ruge et al., 2011a*; *Ruge et al., 2011b*). This contrasts with PD, where the maximal clinical response to DBS occurs within a much faster time course (*Stefani et al., 2019*; *Hammond et al., 2007*). The present study shows disease-specific differences in the dynamics associated with modulation of direct pathway projections, thus, potentially providing a local synaptic rationale for the differential time courses associated with the clinical effects of DBS in the same target. Specifically, we found faster rates of synaptic depression during HFS and greater post-HFS long-term plasticity-like effects in PD compared to dystonia, implying that striato-pallidal synapses in dystonia are perhaps more resistant to change compared to PD. Our findings may therefore explain the phenomenon that functional benefits of DBS require longer time courses in dystonia if the effects of DBS rely upon network reorganization, which involves adjustments in neural connectivity or synaptic efficacy in response to stimulation (*Ruge et al., 2011b*). Indeed, the benefits of GPi-DBS in dystonia have also been shown to persist for long periods of time even after the cessation of stimulation (*Ruge et al., 2011a*). In PD, washout effects take longer with GPi-DBS compared to STN-DBS (*Follett et al., 2010*), and long-term dyskinesia reduction has also been observed with GPi-DBS even when DBS is off (*Anderson et al., 2005*; *Bejjani et al., 2000*). While further work is certainly required to better understand disease-related differences in plasticity, our findings may nevertheless motivate the development of periodic intermittent (ON/OFF) DBS strategies that periodically modulate synaptic plasticity for therapeutic benefits that outlast stimulation delivery as have recently been employed in preclinical work (*Spix et al., 2021*; *Mastro et al., 2017*).

## Limitations and future directions

Important limitations of intraoperative/intracranial studies in humans include a lack of access to healthy control data (hence comparison across disorders), the inability to use pharmacological interventions to verify pathway specificity of elicited responses, and time constraints preventing thorough scrutinization of time courses of long-term plasticity-like effects. Indeed, GPi receives the greatest abundance of inhibitory inputs from striatum (direct pathway), but also it also receives inhibitory inputs by way of GPe (indirect pathway). Although we can functionally disaggregate these pathway-specific responses based on the differences in morphology and dynamics of GPe-mediated versus striatum-mediated inhibitory fEPs (*Steiner et al., 2022*), the possibility of compounded effects cannot be completely ruled out. While our neuronal investigations provide cellular-level support for closed-loop targeting of disease-related neural oscillations (*Johnson et al., 2021*; *Little et al., 2016*), future applications of DBS may also benefit from closed-loop tuning of basal-ganglia-thalamo-cortical circuit dynamics and plasticity through chronic monitoring of evoked potential responses (*Versantvoort et al., 2024*). Indeed, optogenetic studies in Parkinsonian rodents have demonstrated the ability to achieve lasting therapeutic efficacy via periodic activations of striatal direct pathway projections (*Spix et al., 2021*; *Mastro et al., 2017*), likely leveraging long-term plasticity-like mechanisms. An additional limitation of this study is that we did not stringently monitor personal medication intake or levels of intraoperative sedation; however, all patients with PD were asked to withdraw from medications the night before surgery, and the analyses only included patients operated on awake. It is important to consider that classifications of PD and dystonia as hypo- and hyperkinetic disorders can be considered oversimplifications as there can be contradictory comorbidities and drug- and DBS-related effects (*Reese et al., 2015*; *Esposito et al., 2017*), though our results do not change substantially when only hypokinetic PD features are considered (*Figure 1—figure supplement 2*). Additionally, our analyses involved pooling of patients with various forms of dystonia. While we did not observe differences across dystonia subtypes (*Figure 2—figure supplement 1*), future studies in larger patient cohorts are warranted. Finally, as many findings in *Figure 1* do not survive corrections for multiple comparisons, we suggest interpretation of results with caution. Despite this, many of our findings related to neuronal correlates are generally in line with previous literature, especially related to oscillatory correlates of PD and dystonia.

## Conclusion

We substantiated claims of hypo- versus hyperfunctional GPi output in PD and dystonia, while furthermore providing cellular-level validation of the pathological nature of theta and low-beta oscillations in respective disorders. Such circuit changes may be underlain by disease-related differences in plasticity

of striato-pallidal synapses, which are seemingly less plastic and/or respond slower to change in dystonia compared to PD.

## Acknowledgements
The authors thank the patients for their participation.

## Additional information

### Funding

| Funder | Grant reference number | Author |
|---|---|---|
| Brain Canada | | Luka Milosevic |
| Banting Research Foundation | | Luka Milosevic |
| Dystonia Medical Research Foundation Canada | | Luka Milosevic |

The funders had no role in study design, data collection and interpretation, or the decision to submit the work for publication.

### Author contributions
Srdjan Sumarac, Resources, Data curation, Software, Formal analysis, Visualization, Methodology, Writing – original draft; Kiah A Spencer, Data curation, Formal analysis, Investigation, Methodology, Writing – original draft; Leon A Steiner, Conceptualization, Writing – original draft, Writing – review and editing; Conor Fearon, Data curation, Writing – review and editing; Emily A Haniff, Data curation, Writing – original draft; Andrea A Kühn, Project administration, Writing – review and editing; Mojgan Hodaie, Suneil K Kalia, Andres Lozano, Resources, Project administration, Writing – review and editing; Alfonso Fasano, Resources, Data curation, Project administration, Writing – review and editing; William Duncan Hutchison, Conceptualization, Resources, Methodology, Project administration, Writing – review and editing; Luka Milosevic, Conceptualization, Resources, Supervision, Funding acquisition, Validation, Investigation, Methodology, Project administration, Writing – review and editing

### Author ORCIDs
Srdjan Sumarac ⓘ https://orcid.org/0000-0002-3840-5957
William Duncan Hutchison ⓘ https://orcid.org/0000-0002-3866-4940
Luka Milosevic ⓘ https://orcid.org/0000-0002-4051-5397

### Ethics
Human subjects: Each patient provided informed consent and experiments were approved by the University Health Network Research Ethics Board and adhered to the guidelines set by the tri-council Policy on Ethical Conduct for Research Involving Humans.

Reviewer #1 (Public Review): https://doi.org/10.7554/eLife.90454.3.sa1
Reviewer #2 (Public Review): https://doi.org/10.7554/eLife.90454.3.sa2
Author response https://doi.org/10.7554/eLife.90454.3.sa3

## Additional files

### Supplementary files
• Supplementary file 1. Data summary. The table provides a summary of the patient data included in the study. It details the diseases studied, the types of data collected (e.g., neuronal, plasticity), and the corresponding clinical scores.
• Supplementary file 2. Neuronal feature summary. The table compares neuronal features such as

firing rates, patterns, and oscillations between PD and dystonia patients.

• Supplementary file 3. Multiple comparisons. (a) The table provides Bonferroni-corrected statistics for comparing neuronal features across different diseases, and (b) reports the Benjamini–Hochberg false discovery rate-corrected statistics for correlating neuronal features with clinical scores.

• MDAR checklist

## Data availability

The data that support the findings of this study are available in a public repository at https://osf.io/nqzd2/.

The following dataset was generated:

| Author(s) | Year | Dataset title | Dataset URL | Database and Identifier |
|---|---|---|---|---|
| Sumarac S | 2024 | Interrogating basal ganglia circuit function in Parkinson's disease and dystonia | https://osf.io/nqzd2/ | Open Science Framework, nqzd2 |

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
