## [Editor Report · eLife assessment]

This is a **valuable** study of the responses of GPi neurons to deep brain stimulation (DBS) in human Parkinson disease and dystonia patients and finds **convincing** evidence for altered short-term and long-term plasticity in response to DBS between the two patient populations. This dataset is of interest to both basic and clinical researchers working in the field of DBS and movement disorders.

---

## [Referee Report · Reviewer #1 (Public Review)]

Summary:

Sumarac et al investigate differences in globus pallidus internus (GPi) spike activity and short- and long-term plasticity of direct pathway projections in patients with Parkinson's disease (PD) and dystonia. Their main claims are that GPi neurons exhibit distinct characteristics in these two disorders, with PD associated with specific power-frequency oscillations and dystonia showing lower firing rates, increased burstiness, and less regular activity. Additionally, long-term plasticity and synaptic depression appear to differ between the two conditions. The authors suggest that these findings support the concept of hyperfunctional GPi output in PD and hypofunctional output in dystonia, possibly driven by variations in plasticity of striato-pallidal synapses. Overall enthusiasm is relatively high, but I think the discussion omits discussing findings that don't align well with standard models.

Strengths:

- These types of studies are valuable as the data arise from patients who have dystonia or PD. This could provide unique insights into disease pathophysiology that might not be recapitulated in animal systems work.

Comments on latest version:

The authors addressed my concerns in their revision.

---

## [Referee Report · Reviewer #2 (Public Review)]

Summary:

The authors investigated how neuronal activity and metrics of plasticity using local electrical stimulation in the GPi were different between Parkinson's disease and dystonia patients.

Strengths:

The authors achieved their aims of comparing the dynamics related to stimulation induced metrics of plasticity in GPi between dystonia and PD, which has not been previously explored. These results could directly inform DBS protocols to improve treatment. The methods are clearly described and results are strong with measurements from a large population of patients for each disease group, and with distinct findings for each group. These results also may help provide insight as to the differences in terms of dynamics of therapeutic stimulation effects in the different disease groups.

Weaknesses:

After the revisions, the discussion contains many more details and comparisons to relevant literature, which will be helpful for readers to appreciate the importance of the results. The conclusion could have been strengthened as well, as it seems to be a very general summary of their findings without consideration of their clinical implications and importance. However, this may be a minor issue.

---

## [Author Response]

The following is the authors’ response to the original reviews.

**eLife assessment:**
This manuscript is a valuable study of the responses of GPi neurons to DBS stimulation in human PD and dystonia patients and it finds evidence for altered short-term and long-term plasticity in response to DBS between the two patient populations. This data set is of interest to both basic and clinical researchers working in the field of DBS and movement disorders. While there was enthusiasm for the potential significance of these findings, support for their conclusions was incomplete. Thir data may be indicative of more interesting and complex interpretations than currently considered in the article.

The authors would like to express their gratitude to the Editorial Team and Reviewers for their invaluable feedback which helped to improve the manuscript.

**Reviewer #1:**
Summary:Sumarac et al investigate differences in globus pallidus internus (GPi) spike activity and short- and long-term plasticity of direct pathway projections in patients with Parkinson's disease (PD) and dystonia. Their main claims are that GPi neurons exhibit distinct characteristics in these two disorders, with PD associated with specific power-frequency oscillations and dystonia showing lower firing rates, increased burstiness, and less regular activity. Additionally, long-term plasticity and synaptic depression appear to differ between the two conditions. The authors suggest that these findings support the concept of hyperfunctional GPi output in PD and hypofunctional output in dystonia, possibly driven by variations in the plasticity of striato-pallidal synapses. Overall enthusiasm is relatively high, but I think the discussion omits discussing findings that don't align well with standard models.Strengths:These types of studies are valuable as the data arise from patients who have dystonia or PD. This could provide unique insights into disease pathophysiology that might not be recapitulated in animal systems work.

Thank you for the positive feedback.

Weaknesses:- The rate model and indirect/direct pathway ideas lack explanatory power; too much of the hypothesis generation and discussion in this manuscript is set in the context of these old ideas. Their data in my view emphasize this somewhat emphatically. Most patients with the 'hypokinetic' movement disorder PD have dystonia as a part of their motor features. Dystonia is a form of excessive muscle activation that on the one hand is 'hyperkinetic' but on the other usually decreases the speed of motor tasks, even in patients with primary dystonia. Similarly, PD patients display a bewildering variety of hyperkinetic manifestations as well (rest tremor, dystonia, dyskinesia). If these are truly independent classifications, i.e. hyper- versus hypo-kinetic, the authors must acknowledge that there is considerable overlap in the spike activity across groups - numerous dystonia patients display higher discharge rates than the majority of the PD sample. Based on the firing rate alone, it would not be possible to distinguish these groups.

Thank you for your insightful comments regarding the discussion of the rate model and the distinction between hyperkinetic and hypokinetic movement disorders. We acknowledge that the rate model, primarily derived from limited number of animal subjects [1], may not fully encapsulate the complexities of Parkinson's disease (PD) and dystonia. Our study aimed to validate animal model findings in humans by correlating single-neuron features with disease symptom severity. However, we concur with the Reviewer’s comment regarding the overlapping motor features in hypokinetic and hyperkinetic disorders. We can speculate that the overlap in neuronal properties may be reflected in the overlap of, for example, hyperkinetic features being also present in PD, as suggested by the Reviewer. Per the Reviewer’s request, we have now acknowledged this notion in the manuscript. Interestingly, hypokinetic symptoms have been reported to occur in dystonia in response to GPi-stimulation and have been associated with beta activity in the LFP [2], which reinforces the notion that neural activity may be more related to specific symptoms rather than diseases as a whole. Supplementing our analyses, in addition to total UPDRSIII scores, we have now provided correlations with only hypokinetic (i.e. bradykinesia) subscores of the UPDRSIII to focus on more direct assessment of hypokinetic features in PD versus hyperkinetic features in dystonia. We have updated our methods and results accordingly.

[1] M. R. DeLong, “Primate models of movement disorders of basal ganglia origin.,” Trends Neurosci, vol. 13, no. 7, pp. 281–285, Jul. 1990, doi: 10.1016/0166-2236(90)90110-v.

[2] R. Lofredi et al., “Pallidal Beta Activity Is Linked to Stimulation-Induced Slowness in Dystonia,” Movement Disorders, vol. 38, no. 5, pp. 894–899, 2023, doi: 10.1002/mds.29347.

Amendments to the manuscript:

“Indeed, variability in spike firing rates in PD may be reflected in the considerable overlap in spiking activity between PD and dystonia (Fig. 1A), with many dystonia patients exhibiting higher discharge rates compared to PD patients.”

“Given that UPDRSIII includes both hypokinetic and hyperkinetic symptoms of PD, we further sought to disaggregate the score by only considering items 23-26 in UPDRSIII, which assess hypokinetic symptoms of PD.”

“… with a marginally stronger correlation for PD hypokinetic symptoms only (items 23-26 of UPDRSIII, Spearman's rho=0.32, p=.0330; Supplementary Fig. 3)”

Supplementary Fig. 3: We provided correlations with hypokinetic (i.e., bradykinesia) subscore of the UPDRSIII. There is very little difference between correlation results of UPDRSIII total (Fig. 1) and the hypokinetic-only subscore (Supplementary Fig. 3).

“though our results do not change substantially when only hypokinetic PD features are considered (Supplementary Fig. 3).”

- If beta power is pathognomonic of parkinsonism, the authors found no differences in beta-related spike discharges across the groups. One would have predicted greater beta power in PD than in primary dystonia. This should be discussed explicitly and an interpretation should be provided.

We agree with the reviewer that considering the previous LFP literature, one might have expected a difference in single-neuron oscillation power between PD and dystonia. However, while prior studies [3], [4] have reported significant differences in oscillatory power between the two diseases, researchers examined local field potential (LFP) activity only. Other work [5] in non-human primates investigated single-neuron oscillations and reported no differences between PD and dystonia at the single-neuron level, in line with our findings. However, despite the lack of difference in overall power presented here, we provide evidence that the strength of the beta-frequency single-neuron oscillations nevertheless correlates with symptom severity in PD but not dystonia; whereas the strength of the theta-frequency single-neuron oscillations correlates with symptom severity in dystonia but not PD.

[3] P. Silberstein et al., “Patterning of globus pallidus local field potentials differs between Parkinson’s disease and dystonia.,” Brain, vol. 126, no. Pt 12, pp. 2597–2608, Dec. 2003, doi: 10.1093/brain/awg267.

[4] D. D. Wang et al., “Pallidal Deep-Brain Stimulation Disrupts Pallidal Beta Oscillations and Coherence with Primary Motor Cortex in Parkinson’s Disease,” J Neurosci, vol. 38, no. 19, pp. 4556–4568, May 2018, doi: 10.1523/JNEUROSCI.0431-18.2018.

[5] P. A. Starr et al., “Spontaneous pallidal neuronal activity in human dystonia: comparison with Parkinson’s disease and normal macaque.,” J Neurophysiol, vol. 93, no. 6, pp. 3165–3176, Jun. 2005, doi: 10.1152/jn.00971.2004.

Amendments to the manuscript:

“Although previous research has reported differences in the LFP power between PD and dystonia [27,28], a study in non-human primates found no such differences in single-neuron oscillatory strength [8], as reflected in our findings. However, despite a lack of difference in overall power across disorders, we were able to derive disease/frequency-specific relationships with respect to clinical scores (Fig. 1C; oscillatory features).”

- The study lacks a healthy control group, making it challenging to differentiate disease-specific findings from normal variations in GPi activity and plasticity. Although this is acknowledged in the discussion, this complicates the interpretation of the results. The sample sizes for PD and dystonia patients are relatively small, and the study combines various forms of dystonia, potentially masking subtype-specific differences. A larger and more homogenous sample could enhance the study's reliability.

Indeed, intraoperative microelectrode recordings cannot be obtained in healthy individuals. We agree with the Reviewer that this limits the interpretation of the data. However, directly comparing clinical correlations with single neuron readouts between two distinct clinical entities may, to some degree, compensate for the lack of healthy control data. This contrast, while not providing a healthy control, is still able to point to disease-specific differences. This approach has previously been used to comparisons at the LFP level [6]. While the sample size is indeed small, it is comparable or even higher to similar studies that have investigated the relation of symptom severity of single neuron readouts [7]. The Reviewer is right in that we do not differentiate between generalized or cervical dystonia. We chose to do so because our subgroup analysis provided in the Supplementary Material did not suggest specific differences; though there is insufficient data from specific dystonia subtypes to make formal statistical comparisons. Indeed, future studies should investigate specific subtypes further.

[6] R. Lofredi et al., “Pallidal beta bursts in Parkinson’s disease and dystonia,” Movement Disorders, vol. 34, no. 3, pp. 420–424, 2019, doi: 10.1002/mds.27524.

[7] A. Gulberti et al., “Subthalamic and nigral neurons are differentially modulated during parkinsonian gait,” Brain, p. awad006, Feb. 2023, doi: 10.1093/brain/awad006.

Amendments to the manuscript:

“While we did not observe differences across dystonia subtypes (Supplementary Fig. 1), future studies in larger patient cohorts would are warranted. Finally, as many findings in Fig. 1 do not survive corrections for multiple comparisons, we suggest interpretation of results with caution. Despite this, many of our findings related to neuronal correlates are generally in line with previous literature, especially related to oscillatory correlates of PD and dystonia.”

- While they mention that data are available on request, sharing data openly would increase transparency and allow for independent validation of the results. It is unclear how sharing deidentified data would compromise patient privacy or present ethical issues of any kind, as claimed by the authors.

Much of the data in question were collected under an old Research Ethics Board (REB) protocol which did not address data sharing. However, we have consulted with our REB and gained retroactive permission to post de-identified data which are now available in the Supplementary Material.

Amendments to the manuscript:

“The data that support the findings of this study are available in a public repository (see: https://osf.io/nqzd2/)”

- They appropriately acknowledge several limitations, such as the inability to use pharmacological interventions and the need for further research in the chronic setting.

Thank you for the comment.

- The manuscript highlights differences in GPi activity and plasticity between PD and dystonia but could provide more context on the clinical implications of these findings, particularly regarding what the implications would be novel paradigms for deep brain stimulation.

Thank you for the comment. Our finding that striato-pallidal plasticity decays more slowly in dystonia compared to PD may relate to the slower time course of symptom relief associated with GPi-DBS in dystonia, as presently outlined in the discussion. On the other hand, symptoms are also suppressed for longer after the cessation of stimulation in dystonia compared to PD, which may reflect long-term plastic changes [8], [9]. In the context of clinical DBS, plasticity modulation may be facilitated by intermittent stimulation algorithms that may achieve the necessary plastic network change by applying stimulation for a defined time but could then be switched off for improved energy consumption and perhaps as a means of mitigating side effects. DBS devices with chronic sensing may enable monitoring of evoked potential amplitudes for future adaptive stimulation applications; however, currently available devices are limited by low sampling rates, but future devices may overcome these technical limitations.

[8] D. Ruge et al., “Deep brain stimulation effects in dystonia: time course of electrophysiological changes in early treatment.,” Mov Disord, vol. 26, no. 10, pp. 1913–1921, Aug. 2011, doi: 10.1002/mds.23731.

[9] D. Ruge et al., “Shaping reversibility? Long-term deep brain stimulation in dystonia: the relationship between effects on electrophysiology and clinical symptoms.,” Brain, vol. 134, no. Pt 7, pp. 2106–2115, Jul. 2011, doi: 10.1093/brain/awr122.

Amendments to the manuscript:

“While further work is certainly required to better understand disease-related differences in plasticity, our findings may nevertheless motivate the development of periodic intermittent (ON/OFF) DBS strategies which periodically modulate synaptic plasticity for therapeutic benefits which outlast stimulation delivery, as have recently been employed in preclinical work [52,53].”

- While statistical tests are mentioned, the manuscript could benefit from a more detailed presentation of statistical methods, including correction for multiple comparisons and effect sizes. Did the authors consider different recording sites within each patient as independent observations? I think this is not appropriate if that was the case.

Thank you for your constructive feedback. In response to the concerns regarding the statistical methods, we have expanded our analysis to provide a more comprehensive statistical overview. Specifically, we implemented the Bonferroni correction for multiple comparisons across each of the seven tests conducted for the differences in single-neuron features between PD and dystonia. The adjustment revealed that only the burst index and coefficient of variation retain statistical significance after post hoc correction, while the firing rate does not. Results of the Bonferroni corrections are now presented in Supplementary Table 3. Reflecting on the initial comment about firing rates between the two disorders, our updated findings underscore the limitation of using firing rates alone to differentiate between PD and dystonia, and instead, our analysis now points to burstiness and firing irregularity as more reliable discriminators. Regarding the clinical correlations, we refined our statistical analysis by employing nonparametric Monte Carlo permutation tests with 5000 permutations, as used in recent work [10], [11]. This method is chosen for its independence from assumptions regarding data distribution. Specifically, we computed and tested the Spearman rho for significance using the permutation test. Then, to address multiple comparisons, we controlled the false discovery rate (FDR) using the Benjamini-Hochberg procedure. Results of these comparisons are now presented in Supplementary Table 4. Lastly, to address the concern regarding recording site independence within patients, we updated our plasticity analysis methodology. In our study, 6 out of 18 patients had multiple recording sites. Thus, to account for this, we employed linear mixed models (LMM) with patient ID as a random factor to appropriately account for the non-independence of these observations.

[10] v Lofredi et al., “Dopamine-dependent scaling of subthalamic gamma bursts with movement velocity in patients with Parkinson’s disease,” Elife, vol. 7, p. e31895, Feb. 2018, doi: 10.7554/eLife.31895.

[11] R. Lofredi et al., “Subthalamic beta bursts correlate with dopamine-dependent motor symptoms in 106 Parkinson’s patients,” npj Parkinsons Dis., vol. 9, no. 1, Art. no. 1, Jan. 2023, doi: 10.1038/s41531-022-00443-3.

Amendments to the manuscript:

“For comparing differences in single-neuron features between PD and dystonia, significant results were followed up with post hoc multiple comparisons with a Bonferroni correction. For clinical correlations, non-parametric Monte Carlo permutation tests were used, avoiding assumptions about data distribution. The tested values were randomly shuffled 5,000 times to form a probability distribution, with the p-value reflecting the original sample rank. All tests underwent adjustment for multiple comparisons, controlling the false discovery rate (FDR) at an α-level of 0.05.”

“analyzed using a linear mixed model (LMM) with patient ID as a random factor, normalized fEP amplitudes as the response variable, and epoch as a fixed effect”

“using a LMM with patient ID as a random factor”

“However, none of the clinical correlations survived Benjamini-Hochberg FDR-correction for multiple comparisons (Supplementary Table 4).”

“In PD, fEP amplitudes were significantly greater after compared to before HFS (LMM; p = .0075, effect size = 5.42 ± 1.79; Fig. 2C), while in dystonia, the increase approached but did not reach statistical significance (LMM; p = .0708, effect size = 2.82 ± 1.45; Fig. 2C).”

All statistics were updated in the results section and the figures.

“Finally, as many findings in Fig. 1 do not survive corrections for multiple comparisons, we suggest interpretation of results with caution. Despite this, many of our findings related to neuronal correlates are generally in line with previous literature, especially related to oscillatory correlates of PD and dystonia.”

- The manuscript could elaborate on the potential mechanisms underlying the observed differences in GPi activity and plasticity and their relevance to the pathophysiology of PD and dystonia.

Thank you for your feedback. We have enhanced the manuscript by integrating additional discussions on previous studies related to plasticity in dystonia and PD (e.g., [12], [13]), which highlight excessive plasticity in dystonia. Although these may appear contradictory to our findings of increased plasticity in PD compared to dystonia, we propose (also justified by previous literature) that chronic dopaminergic medication use may lead to synaptic over-sensitization, which has been hypothesized as a biological mechanism underlying levodopa-induced dyskinesias (a hyperkinetic feature) in PD [14].

[12] Y. Tamura et al., “Disordered plasticity in the primary somatosensory cortex in focal hand dystonia.,” Brain, vol. 132, no. Pt 3, pp. 749–755, Mar. 2009, doi: 10.1093/brain/awn348.

[13] D. A. Peterson, T. J. Sejnowski, and H. Poizner, “Convergent evidence for abnormal striatal synaptic plasticity in dystonia.,” Neurobiol Dis, vol. 37, no. 3, pp. 558–573, Mar. 2010, doi: 10.1016/j.nbd.2009.12.003.

[14] P. Calabresi, B. Picconi, A. Tozzi, V. Ghiglieri, and M. Di Filippo, “Direct and indirect pathways of basal ganglia: a critical reappraisal.,” Nat Neurosci, vol. 17, no. 8, pp. 1022–1030, Aug. 2014, doi: 10.1038/nn.3743.

Amendments to the manuscript:

“Converging evidence from past animal and human studies suggests that dystonia is associated with impaired synaptic function and abnormal synaptic plasticity [35–37]. Compared to healthy controls, it has been shown that transcranial magnetic stimulation induced motor evoked potentials (MEPs) are hyperexcitable in dystonia [38,39], and somatosensory and motor cortical plasticity is greater [40]. Likewise, enhanced long-term potentiation at cortico-striatal synapses has been shown in rodent models of dystonia [41,42]. While our finding that long term potentiation effects are greater in PD compared to dystonia (Fig. 2D) is difficult to corroborate with this literature, one potential explanation can be that all of our PD patients are long-term users of levodopa. We have previously shown that the intake of this antiparkinsonian dopaminergic medication leads to potent increases in the magnitude of direct pathway plasticity [15]. Although patients are 12hr withdrawn form antiparkinsonian medications for surgery, it could be that striato-pallidal synapses are nevertheless chronically over-sensitized from prolonged use of dopaminergic medication; which is a well-known hypothesis related to the manifestation of levodopa-induced dyskinesias (a hyperkinetic feature) in PD [43]. Indeed, a lack of depotentiation of striato-pallidal projections has previously been observed in patients with levodopa-induced dyskinesias [44]. As such, excessive plasticity of these projections may corroborate hyperkinetic features of dystonia and levodopa-induced dyskinesias in PD.”

**Reviewer #2:**
Summary:The authors investigated how neuronal activity and metrics of plasticity using local electrical stimulation in the GPi were different between Parkinson's disease and dystonia patients.Strengths:The introduction highlights the importance of the work and the fundamental background needed to understand the rest of the paper. It also clearly lays out the novelty (i.e., that the dynamics of plastic effects in GPi between dystonia and PD have not been directly compared).The methods are clearly described and the results are well organized in the figures.The results are strong with measurements from a large population of patients for each disease group and with distinct findings for each group.

Thank you for the kind appraisal.

Weaknesses:The discussion was hard to follow in several places, making it difficult to fully appreciate how well the authors' claims and conclusions are justified by their data, mostly in relation to the plasticity results. It may help to summarize the relevant findings for each section first and then further expand on the interpretation, comparison with prior work, and broader significance. Currently, it is hard to follow each section without knowing which results are being discussed until the very end of the section. With the current wording in the "Neuronal correlates.." section, it is not always clear which results are from the current manuscript, and where the authors are referring to past work.

Thank you for this feedback. The main findings are now summarized in a paragraph at the beginning of the Discussion section, before being discussed in comparison to other studies in the literature in subsequent sub-sections. Moreover, throughout the Discussion, findings from our study are now always reflected by a reference to the relevant figure to more easily differentiate current findings from previous literature. Additionally, Discussion sub-sections have been expanded to consider additional literature in response to various comments throughout the Review process (including the subsequent Review comment).

Amendments to the manuscript:

Paper findings are referenced to figures which depict the results at hand; discussion sub-sections expanded; and the following text has been added at the start of the Discussion:

“In particular, we found that GPi neurons exhibited lower firing rates, but greater burstiness and variability in dystonia compared to PD (Fig. 1A). While no differences were found in the power of spiketrain oscillations across disorders (Fig. 1B), we found that PD symptom severity positively correlated with the power of low-beta frequency spiketrain oscillations, whereas dystonia symptom severity positively correlated with the power of theta frequency spiketrain oscillations (Fig. 1C). Dystonia symptom severity moreover correlated negatively with firing rate, and positively with neuronal variability. These results are discussed in greater detail with respect to previous literature in the subsequent Discussion section entitled “Neuronal correlates of PD and dystonia.” In response to electrical stimulation (protocol depicted in Fig. 2A), we found significant increases in the amplitudes of positive-going stimulation-evoked field potential amplitudes (considered to reflect striato-pallidal synaptic strength; as exemplified in Fig. 2B) before versus after HFS in both PD and dystonia (Fig. 2C); with recording sites in PD exhibiting significantly greater increases (Fig. 2D). While changes to evoked potential amplitude before versus after stimulation can be considered to be reflective of long-term plasticity [15,18], the dynamics of evoked potentials during HFS (as depicted in Fig. 2E) can be considered as reflective of short-term synaptic plasticity [18,21]. To this end, our findings are suggestive of faster latency synaptic depression in PD compared to dystonia (Fig. 2F/G). Plasticity findings are discussed in greater detail in the Discussion section entitled “Direct pathway plasticity.”

Also, I felt that more discussion could be used to highlight the significance of the current results by comparing and/or contrasting them to prior relevant work and mechanisms. The novelty or impact is not very clear as written. Could this be further substantiated in the Discussion?

Thank you for the feedback. The discussion has been expanded to include additional literature that is relevant to the findings reported in the manuscript. For example, with regards to the neuronal correlates sub-section, we now highlight the important findings [15] that show changes to the discharge rates and oscillatory tendencies of GPi neurons in non-human primates in response to staged MPTP applications to progressively titrate motor severity; these results substantiate our lack of correlation with firing rates in PD, and presence of a clinical correlation with beta oscillations. We additionally now emphasize human studies that found LFP power difference between PD and dystonia [3], [4]; but simultaneously highlight studies that did not find such differences in spike-train oscillations (in non-human primates) [5], which is reflective of our own findings. With regards to our plasticity sub-section, we have added new content related to previous literature on plasticity in dystonia and PD (also addressed in response to a query from Reviewer #1). For example, we bring to light a variety of previous studies [12], [13] emphasizing excessive plasticity in dystonia. However, while such studies may seem to contradict our findings of greater plasticity in PD compared to dystonia, we additionally provide hypotheses (justified by previous literature) that prolonged used of dopaminergic medication may result in synaptic over-sensitization, thus giving rise to levodopa-induced dyskinesias (a hyperkinetic feature) in PD [14].

[3] P. Silberstein et al., “Patterning of globus pallidus local field potentials differs between Parkinson’s disease and dystonia.,” Brain, vol. 126, no. Pt 12, pp. 2597–2608, Dec. 2003, doi: 10.1093/brain/awg267.

[4] D. D. Wang et al., “Pallidal Deep-Brain Stimulation Disrupts Pallidal Beta Oscillations and Coherence with Primary Motor Cortex in Parkinson’s Disease,” J Neurosci, vol. 38, no. 19, pp. 4556–4568, May 2018, doi: 10.1523/JNEUROSCI.0431-18.2018.

[5] P. A. Starr et al., “Spontaneous pallidal neuronal activity in human dystonia: comparison with Parkinson’s disease and normal macaque.,” J Neurophysiol, vol. 93, no. 6, pp. 3165–3176, Jun. 2005, doi: 10.1152/jn.00971.2004.

[12] Y. Tamura et al., “Disordered plasticity in the primary somatosensory cortex in focal hand dystonia.,” Brain, vol. 132, no. Pt 3, pp. 749–755, Mar. 2009, doi: 10.1093/brain/awn348.

[13] D. A. Peterson, T. J. Sejnowski, and H. Poizner, “Convergent evidence for abnormal striatal synaptic plasticity in dystonia.,” Neurobiol Dis, vol. 37, no. 3, pp. 558–573, Mar. 2010, doi: 10.1016/j.nbd.2009.12.003.

[14] P. Calabresi, B. Picconi, A. Tozzi, V. Ghiglieri, and M. Di Filippo, “Direct and indirect pathways of basal ganglia: a critical reappraisal.,” Nat Neurosci, vol. 17, no. 8, pp. 1022–1030, Aug. 2014, doi: 10.1038/nn.3743.

[15] A. Muralidharan et al., “Physiological changes in the pallidum in a progressive model of Parkinson’s disease: Are oscillations enough?,” Exp Neurol, vol. 279, pp. 187–196, May 2016, doi: 10.1016/j.expneurol.2016.03.002.

Amendments to the manuscript:

“Despite the lack of correlations with firing rate in PD, our findings seem to align with those of Muralidharan and colleagues [25], who showed that GPi neuronal firing rates may not directly correlate with motor severity but exhibit variability across the disease severity continuum in parkinsonian non-human primates (initially increasing, then decreasing, then increasing again at mild, moderate, and severe disease manifestations, respectively). Thus, while GPi discharge rates may change in PD, such changes may not be reflected by linear relationships with motor sign development and progression. Indeed, variability in spike firing rates in PD may be reflected in the considerable overlap in spiking activity between PD and dystonia (Fig. 1A), with many dystonia patients exhibiting higher discharge rates compared to PD patients. While differences in discharge rates were nevertheless observed between PD and dystonia, it may be that the combination of rate and pattern (reflected in the BI and CV) changes best differentiates the two disorders.”

“Converging evidence from past animal and human studies suggests that dystonia is associated with impaired synaptic function and abnormal synaptic plasticity [35–37]. Compared to healthy controls, it has been shown that transcranial magnetic stimulation induced motor evoked potentials (MEPs) are hyperexcitable in dystonia [38,39], and somatosensory and motor cortical plasticity is greater [40]. Likewise, enhanced long-term potentiation (LTP) at cortico-striatal synapses has been shown in rodent models of dystonia [41,42]. While our finding that LTP effects are greater in PD compared to dystonia (Fig. 2D) is difficult to corroborate with this literature, one potential explanation can be that all of our PD patients are long-term users of levodopa. We have previously shown that the intake of this antiparkinsonian dopaminergic medication leads to potent increases in the amount of plasticity elicited in GPi [15]. Although patients are 12hr withdrawn form antiparkinsonian medications for surgery, it could be that striato-pallidal synapses are nevertheless chronically over-sensitized from prolonged use of dopaminergic medication; which is a well-known hypothesis related to the manifestation of levodopa-induced dyskinesias (a hyperkinetic feature) in PD [43]. Indeed, a lack of depotentiation of striato-pallidal projections has previously been observed in patients with levodopa-induced dyskinesias [44]. As such, excessive plasticity of these projections may corroborate hyperkinetic features of dystonia and levodopa-induced dyskinesias in PD.”

Some specific comments and questions about the Discussion:Lines 209-211 - This sentence was hard to understand, could it be clarified?Lines 211-213 - What do phasic and tonic components mean exactly? Could this be specifically defined? Are there specific timescales (as referred to in Intro)?Lines 215-217 - It's not clear what was delayed in dystonia, and how the authors are trying to contrast this with the faster time course in PD. I think some of this is explained in the introduction, but could also be re-summarized here as relevant to the results discussed.Lines 223-224 - I'm not sure I follow the implication that network reorganization leads to delayed functional benefits. Could this be further elaborated?

Reply & Amendments to the manuscript: Thank you for your feedback. We've made the following concise revisions to address the comments:

We've clarified lines 209-211 to explain that variations in electrical stimulation effects on pathways in PD and dystonia may reveal the operational mechanisms of DBS, despite a common target:

“The variation in the modulation of these projections / pathways to electrical stimulation may also indicate the mechanism by which DBS operates across PD and dystonia, despite a common stimulation target.”

In response to the second comment on lines 211-213 about phasic and tonic components, we now specify that phasic refers to dynamic muscle contractions, and tonic to continuous muscle contractions, providing clear definitions relevant to our context:

“Clinical studies in dystonia have shown that DBS leads to a more rapid improvement in the transient, dynamic muscle contractions (phasic components) of the disorder when compared to the sustained, continuous muscle contractions (tonic or fixed components) [33]”

For lines 215-217, we've refined our discussion to clearly contrast the delayed response in dystonia with the faster onset in PD:

“This contrast with PD, where the, the maximal clinical response to DBS occurs within a much faster time course [13,36].”

On lines 223-224, we've expanded the explanation of how network reorganization may lead to delayed functional benefits, highlighting adjustments in neural connectivity and synaptic efficacy in response to stimulation:

“which involves adjustments in neural connectivity or synaptic efficacy in response to the stimulation [14,35].”

Could the absence of a relationship between FR and disease in PD be discussed?

Thank you for raising this point. Despite observing higher firing rates in PD compared to dystonia, it is unexpected that these rates do not correlate with symptom severity according to the rate model of PD [1]. However, despite the lack of correlations with firing rates, our findings align with similar animal work of Muralidharan et al. [15], which reported that neuronal firing rates within the GPi of rhesus monkeys did not increase linearly with respect to varying intensities of parkinsonian motor severity. We did however show that low beta oscillatory strength within the GPi may play a significant role in the manifestation of motor symptoms in PD; which is also in line with findings of Muralidharan and colleagues. As per the Reviewer’s request, we have included this content into our discussion.

[1] M. R. DeLong, “Primate models of movement disorders of basal ganglia origin.,” Trends Neurosci, vol. 13, no. 7, pp. 281–285, Jul. 1990, doi: 10.1016/0166-2236(90)90110-v.

[15] A. Muralidharan et al., “Physiological changes in the pallidum in a progressive model of Parkinson’s disease: Are oscillations enough?,” Exp Neurol, vol. 279, pp. 187–196, May 2016, doi: 10.1016/j.expneurol.2016.03.002.

Amendments to the manuscript:

“Despite the lack of correlations with firing rate in PD, our findings seem to align with those of Muralidharan and colleagues [25], who showed that GPi neuronal firing rates may not directly correlate with motor severity but exhibit variability across the disease severity continuum in parkinsonian non-human primates (initially increasing, then decreasing, then increasing again at mild, moderate, and severe disease manifestations, respectively). Thus, while GPi discharge rates may change in PD, such changes may not be reflected by linear relationships with motor sign development and progression.”

“Indeed, Muralidharan and colleagues [25] also showed linear group-level relationships between low-beta frequency spiketrain oscillations and disease severity in parkinsonian non-human primates, despite the lack of linear relationships with spike discharge rates (as discussed above).”

It wasn't very clear how the direct pathway can be attributed to plasticity changes if the GPi makes up both the direct and indirect pathways. Could this be further clarified?

The reviewer brings up an important nuanced point. Recent work from our lab [16] shows that inhibitory evoked fields in STN (which receives inhibitory fields from GPe; no other inhibitory sources) are persistent with very minimal depression during HFS. On the other hand, inhibitory fields in the SNr (which receives majority of its inhibitory inputs from striatum; though some come by way of GPe as well per anatomical literature) depress quickly. We have previously also shown these rapidly depressing fields in GPi [17], [18], which also receives the majority of its inhibitory inputs via striatum, though some also from GPe. As such, the disaggregation of striatum-mediated versus GPe-mediated inhibitory fields is achieved based on: lack of rapidly depressing inhibitory evoked field potentials in STN (which receives inhibitory inputs via GPe and not striatum), but a common presence of rapidly depressing evoked field potentials in SNr and GPi (which both receive most of their inhibitory inputs from striatum); differences in the morphology of purportedly GPe- (fast latency) versus striatum-mediated (slow latency) evoked field potentials [16]; and the presence of slow latency caudato-nigral evoked field potentials in slices [19] that are reversed by GABA antagonist application [20]. These points are indeed outlined in the first paragraph of the Discussion sub-section “Direct pathway plasticity.” However, we have now additionally added a point to the Limitations that inhibitory inputs to the GPi also come by way of GPe, though in a lesser abundance.

[16] L. A. Steiner et al., “Persistent synaptic inhibition of the subthalamic nucleus by high frequency stimulation,” Brain Stimul, vol. 15, no. 5, pp. 1223–1232, 2022, doi: 10.1016/j.brs.2022.08.020.

[17] L. D. Liu, I. A. Prescott, J. O. Dostrovsky, M. Hodaie, A. M. Lozano, and W. D. Hutchison, “Frequency-dependent effects of electrical stimulation in the globus pallidus of dystonia patients.,” J Neurophysiol, vol. 108, no. 1, pp. 5–17, Jul. 2012, doi: 10.1152/jn.00527.2011.

[18] L. Milosevic et al., “Modulation of inhibitory plasticity in basal ganglia output nuclei of patients with Parkinson’s disease,” Neurobiology of Disease, vol. 124, pp. 46–56, Apr. 2019, doi: 10.1016/j.nbd.2018.10.020.

[19] M. Yoshida and W. Precht, “Monosynaptic inhibition of neurons of the substantia nigra by caudato-nigral fibers,” Brain Res, vol. 32, no. 1, pp. 225–228, Sep. 1971, doi: 10.1016/0006-8993(71)90170-3.

[20] W. Precht and M. Yoshida, “Blockage of caudate-evoked inhibition of neurons in the substantia nigra by picrotoxin,” Brain Res, vol. 32, no. 1, pp. 229–233, Sep. 1971, doi: 10.1016/0006-8993(71)90171-5.

Amendments to the manuscript:

“Indeed, GPi receives the greatest abundance of inhibitory inputs from striatum (direct pathway), but also it also receives inhibitory inputs by way of GPe (indirect pathway). Although we can functionally disaggregate these pathway-specific responses based on differences in morphology and dynamics of GPe-mediated versus striatum-mediated inhibitory fEPs [21]; the possibility of compounded effects cannot be completely ruled out.”

The mechanism of short- and long-term plasticity as applied in the protocols used in this work are outlined in reference to previous citations [15, 16, 18]. Because this is a central aspect of the current work and interpreting the results, it was difficult to appreciate how these protocols provide distinct metrics of short and long-term plasticity in GPi without some explanation of how it applies to the current work and the specific mechanisms. It would also help to be able to better link how the results fit with the broader conclusions.

Short-term plasticity is measured as the dynamic change to the fEP during ongoing HFS. For long-term plasticity analyses, the fEP amplitudes during LFS were compared pre- versus post-HFS. To make this analysis more intuitive we have added a protocol illustration to Fig 2. We have moreover greatly expanded the discussion to include more literature related to disease-specific differences in plasticity, and implications of modulating plasticity using DBS.

Amendments to the manuscript:

Added new panel to Fig 2

“Converging evidence from past animal and human studies suggests that dystonia is associated with impaired synaptic function and abnormal synaptic plasticity [35–37]. Compared to healthy controls, it has been shown that transcranial magnetic stimulation induced motor evoked potentials (MEPs) are hyperexcitable in dystonia [38,39], and somatosensory and motor cortical plasticity is greater [40]. Likewise, enhanced long-term potentiation at cortico-striatal synapses has been shown in rodent models of dystonia [41,42]. While our finding that long term potentiation effects are greater in PD compared to dystonia (Fig. 2D) is difficult to corroborate with this literature, one potential explanation can be that all of our PD patients are long-term users of levodopa. We have previously shown that the intake of this antiparkinsonian dopaminergic medication leads to potent increases in the amount of plasticity elicited in GPi [15]. Although patients are 12hr withdrawn form antiparkinsonian medications for surgery, it could be that striato-pallidal synapses are nevertheless chronically over-sensitized from prolonged use of dopaminergic medication; which is a well-known hypothesis related to the manifestation of levodopa-induced dyskinesias (a hyperkinetic feature) in PD [43]. Indeed, a lack of depotentiation of striato-pallidal projections has previously been observed in patients with levodopa-induced dyskinesias [44]. As such, excessive plasticity of these projections may corroborate hyperkinetic features of dystonia and levodopa-induced dyskinesias in PD.”

In the Conclusion, it was difficult to understand the sentence about microcircuit interaction (line 232) and how it selectively modulates the efficacy of target synapses. Some further explanation here would be helpful. Also, it was not clear how these investigations (line 237) provide cellular-level support for closed-loop targeting. Could the reference to closed-loop targeting also be further explained?

We agree with the reviewer that the current wording may be confusing. We have changed the wording to be clearer. We have additionally added content related to closed-loop DBS based on chronic monitoring of evoked potential responses.

Amendments to the manuscript:

“Furthermore, chronic monitoring of evoked fields may allow for tracking of subcortical neuronal projections as indexed by inhibitory fields reported in this study. microcircuit interaction to selectively modulate the efficacy of target synapses.”

future applications of DBS may also benefit from closed loop tuning of basal-ganglia-thalamo-cortical circuit dynamics and plasticity through chronic monitoring of evoked potential responses [56].

How is the burst index calculated (Methods)?

Thank you for pointing out that the burst index definition was missing from the paper. It has now been added to the manuscript.

Amendments to the manuscript:

“The burst index was computed by taking the ratio of the means from a two-component Gaussian mixture model applied to the log interspike interval distribution, a modification of the previous mode-over-mean ISI method [20]”

Figures and figure captions are missing some details:Fig. 1 - What does shading represent?

The shading in Fig. 1 illustrates results that were significant before adjustment for multiple comparisons.

Amendments to the manuscript:

“Depicted scatterplots are results that were significant before correction for multiple comparisons”

Fig. 2 - Can the stimulation artifact be labeled so as not to be confused with the physiological signal? Is A representing the average of all patients or just one example? Are there confidence intervals for this data as it's not clear if the curves are significantly different or not (may not be important to show if just one example)? Same for D. What is being plotted in E? Is this the exponential fitted on data? Can this be stated in the figure citation directly so readers don't have to find it in the text, where it may not be directly obvious which figure the analyses are being applied towards?

Thank you for your comments regarding Fig. 2. We have made the following revisions to address the concerns:

To clarify the presence of stimulation artifacts and differentiate them from the physiological signal, we have updated Panel B and E in the updated Fig. 2 which highlight the stimulation artifacts accordingly.

Regarding the comment about Panel A (now B in the updated figure), it represents one single example per disease, rather than an average of all patients.

In response to the comment about what is plotted in Panel E, we have revised the figure caption to explicitly state that it includes the exponential fit on the data.

Amendments to the manuscript:

Figure 2 panel B and E now highlight stimulation artifacts.

**Author response image 2. sa3fig2:** 

**Author response image 3. sa3fig3:** 

The figure captions could use more details, that can be taken from the text, so that readers can understand figures without searching for relevant details across the paper.

Thank you for your feedback. We have revised the figure captions accordingly to provide more details.

Amendments to the manuscript:

“Fig 1 – GPi spiketrain feature analyses and clinical correlates of PD and dystonia. (A) With respect to (A) rate-based spiketrain features, firing rate was greater in PD while burst index (BI) and coefficient of variation (CV) were greater in dystonia; whereas no differences were found for (B) oscillatory spiketrain features for theta, alpha, low beta, high beta frequencies. MWU statistical results depicted are not corrected for multiple comparisons; after correction using the Bonferroni method, only CV and BI results remain significant (please see Supplementary Table 3). (C) In PD, the power of low beta spiketrain oscillations positively correlated (Spearman correlation) with symptom severity; in dystonia, neuronal firing rate negatively correlated with symptom severity, whereas CV and the power of theta spiketrain oscillations positively correlated with symptom severity. Depicted scatterplots are results that were significant before correction for multiple comparisons; however, none of the results persist after Benjamini-Hochberg correction for false discovery rate (please see Supplementary Table 4).”

“Fig 2 – Long-term and short-term effects of HFS on striato-pallidal plasticity in PD and dystonia. (A) Schematic of the plasticity protocol to assess long-term plasticity via fEP amplitude comparisons pre- versus post-HFS and short-term plasticity via fEP dynamics during HFS. (B) Highlights example fEP traces for measuring long-term plasticity pre- versus post-HFS, with (C) displaying group-level fEP amplitudes pre- versus post-HFS across diseases. (D) Illustrates the amount of plasticity (i.e., percentage change in fEP amplitudes pre- versus post-HFS) in both PD and dystonia, with PD showing higher levels of plasticity. (E) Provides an example of fEP traces during HFS for assessing short-term plasticity, with (F) depicting group-level decay rates of fEP amplitudes using an exponential fit on the fEP amplitudes over the first 5 stimulus pulses across diseases. (G) Shows the half-life of the fitted exponential (i.e., rate of attenuation of fEP amplitudes) between PD and dystonia, with PD demonstrating faster fEP attenuation.”